# Zinc-Doped Magnesium Borate Glass: A Potential Thermoluminescence Dosimeter for Extended Range of Dosimetric Applications

Muhammad Bakhsh [1,2,3,*], Hiroshi Yasuda [4,*], Nisar Ahmad [5], Jeannie Hsiu Ding Wong [2] and Iskandar Shahrim Mustafa [1]

1   School of Physics, Universiti Sains Malaysia, Penang 11800, Malaysia; iskandarshah@usm.my
2   Department of Biomedical Imaging, Faculty of Medicine, University of Malaya, Kuala Lumpur 50603, Malaysia; jeannie.wong@ummc.edu.my
3   Department of Medical Physics, Karachi Institute of Radiotherapy and Nuclear Medicine, Karachi 11800, Pakistan
4   Department of Radiation Biophysics, Research Institute for Radiation Biology and Medicine, Hiroshima University, Kasumi 1-2-3, Minami-ku, Hiroshima 734-8553, Japan
5   Department of Physics, Balochistan University of Information Technology, Engineering and Management Sciences, Quetta 87100, Pakistan; ahmadnisar31@gmail.com
*   Correspondence: mbnizamani@hotmail.com (M.B.); hyasuda@hiroshima-u.ac.jp (H.Y.)

**Featured Application: The original glass material synthesized in this study is expected to be functional and practical in various applications.**

**Abstract:** In this study, we report the dosimetric properties of zinc-doped magnesium borate ($MgB_4O_7$:Zn) glass, which was originally synthesized. $MgB_4O_7$:Zn glass was successfully synthesized through the melt-quenching technique. The amorphous nature of the synthesized samples was observed through X-ray diffraction (XRD) analysis and further confirmed through field emission scanning electron microscopy (FESEM) analysis. The glass-forming ability and thermal stability were estimated to be 0.61 and 1.62, respectively. The TL dosimetric characteristics, i.e., dose response, reproducibility, TL sensitivity, minimum detectable dose and signal stability, are reported. The synthesized sample demonstrated a simple glow curve with a single well-defined dosimetric peak at 240 °C with an optimal heating rate of 7 °C s$^{-1}$. The synthesized glass demonstrated a linear dose response from 3 Gy to 5 kGy. The promising dosimetric characteristics demonstrate the potential of the synthesized glass to be recommended as a TL dosimeter for a wide range of applications.

**Keywords:** magnesium borate glass; zinc; dopant; thermoluminescence; dosimetry; γ-rays

## 1. Introduction

The thermoluminescence (TL) dosimetry technique is a well-established radiation dosimetry technique. However, despite the range of commercially available TL dosimeters, the need for a single robust, tissue-equivalent and highly sensitive thermoluminescence dosimeter with excellent signal stability is always desirable for an extended range of dosimetric applications. The development of a highly sensitive tissue-equivalent thermoluminescence dosimeter with excellent signal stability remains a challenge for researchers. Researchers, over the years, have laboriously explored different compounds to develop a single TL dosimeter for a wide range of dosimetric applications.

Magnesium borate ($MgB_4O_7$) remains a host matrix of interest due to its unique dosimetric characteristics and has been extensively explored since it was first developed 50 years ago [1–10]. Most of the literature focused on the effectiveness of rare earth dopants [1,2,6–8,11–17]. While the dosimetric properties were improved to some extent with the introduction of various dopants, the crystalline nature of the reported materials

restricted the user due to the need for careful handling of the material [18]. The low effective atomic number ($Z_{eff}$ = 8.4), which is close to human muscle ($Z_{eff}$ = 7.64), makes the $MgB_4O_7$-based thermoluminescence dosimeters suitable for clinical radiation dosimetry, whereas the high neutron capture cross-section of $^{10}B$ attracts researchers to develop a $MgB_4O_7$-based neutron dosimeter [8,19,20]. The optical fading of these detectors remains a serious concern that affects the signal stability significantly and restricts the material to limited dosimetric applications. The suitability of glass-based luminescence dosimeters in postal dosimetric audits is well recognized, thus attracting attention and more focus on glass-based luminescence dosimeters [21,22].

In this study, we report the synthesis, structural and dosimetric properties of a novel $MgB_4O_7$:Zn glass-based TLD. To the best of the authors' knowledge, this is the first report of the use of this material for radiation dosimetry.

## 2. Materials and Methods

### 2.1. Sample Preparation and Characterization

The $MgB_4O_7$:Zn glass series was made from zinc oxide (ZnO), magnesium oxide (MgO) and boron oxide ($B_2O_3$) in powder form, with >99% analytical-grade purity (Alfa Aser, Haverhill, MA, USA). The nominal compositions $(35-Y)MgO-65B_2O_3$, with $Y(ZnO) = 0.1 \leq Y \leq 1$, were prepared using the melt-quenching technique [23–25]. The chemical composition of the individual glass samples is shown in Table 1. The detailed preparation process can be found in the literature [26].

**Table 1.** Composition of synthesized samples.

| Sample ID | Composition (mol%) | | |
| --- | --- | --- | --- |
| | $B_2O_3$ | MgO | ZnO |
| | Host | Modifier | Dopant |
| S35651 | 65.00 | 34.90 | 0.10 |
| S35652 | 65.00 | 34.80 | 0.20 |
| S35653 | 65.00 | 34.70 | 0.30 |
| S35654 | 65.00 | 34.60 | 0.40 |
| S35655 | 65.00 | 34.50 | 0.50 |
| S35656 | 65.00 | 34.40 | 0.60 |
| S35657 | 65.00 | 34.30 | 0.70 |
| S35658 | 65.00 | 34.20 | 0.80 |
| S35659 | 65.00 | 34.10 | 0.90 |
| S356510 | 65.00 | 34.00 | 1.00 |

The structural and thermal characteristics of the synthesized samples were observed through X-ray diffraction, field emission scanning electron microscopy (FESEM), energy dispersive X-ray (EDX) spectroscopy and differential scanning calorimetry (DSC). The amorphous nature of the synthesized samples was confirmed through X-ray diffraction (XRD) analysis using a D8 ADVANCE diffractometer (BRUKER corporation, Billerica, MA, USA). The XRD profiles were collected for powdered glass samples at 2θ from 20° to 80° at a scanning rate of 0.02° s$^{-1}$. The phase homogeneity and surface morphology of the synthesized samples were analyzed through FESEM and elemental composition was confirmed through EDX spectroscopy using FEI NOVA NANOSEM 450 (FEI, Hillsboro, OR, USA). The thermal stability of synthesized glasses and the glass-forming ability of the proposed mixtures were estimated through DSC using SETSYS Evolution (SETARAM Instrumentation, Caluire-et-Cuire, France). The DSC thermogram was obtained between the temperature range of 50 °C to 1000 °C with a heating rate of 10 °C/min. The temperature accuracy of the SETSYS Evolution is ±0.1 °C [27].

*2.2. Dosimetric Characterization*

The dosimetric evaluation of the synthesized glass series was carried out at the Malaysian Nuclear Agency. The synthesized glasses were irradiated to γ-rays from a Cs-137 source. Subsequent thermoluminescence measurements were performed using Harshaw TLD$^{TM}$ reader model 3500, after 24 h, to eliminate the low-temperature peaks associated with shallow traps, which contribute to higher signal instability.

### 2.2.1. Heating Rate Optimization

A group of samples was identified, encapsulated and labeled. The samples were irradiated to a test dose of 10 Gy under standard irradiation conditions. Thermoluminescence measurements were obtained after 24 h to eliminate the low-temperature peaks associated with shallow traps, using different heating rates ranging from 1 °C s$^{-1}$ to 15 °C s$^{-1}$ to identify practically and economically effective heating rates.

### 2.2.2. Composition Optimization

A group of samples from each batch was identified, encapsulated and labeled. The samples were irradiated to a test dose of 10 Gy under standard irradiation conditions. Thermoluminescence measurements were obtained after 24 h to eliminate the low-temperature peaks associated with shallow traps, using the identified optimal heating.

### 2.2.3. Annealing Process

A group of samples with optimal composition was identified, encapsulated and labeled. The samples were irradiated to a test dose of 10 Gy under standard irradiation conditions. Thermoluminescence measurements were obtained after 24 h to eliminate the low-temperature peaks associated with shallow traps, using the optimal heating rate. The annealing process consists of two components: annealing time and annealing temperature. Therefore, in the first step, the 30 min annealing time was fixed and the annealing temperature varied from 100 °C to 400 °C in 50 °C increments to identify the optimum annealing temperature. In the second step, the annealing temperature was fixed at a pre-identified annealing temperature and the annealing time varied from 15 min to 180 min to identify the optimum annealing time.

### 2.2.4. Dose Response

A group of dosimeters was identified, annealed, encapsulated and labeled. The samples were irradiated to pre-identified test doses, ranging from 3 Gy to 5 kGy, under standard irradiation conditions using a Cs-137 gamma cell. Thermoluminescence measurements were obtained after 24 h to eliminate the low-temperature peaks associated with shallow traps using the optimal heating rate.

### 2.2.5. Reproducibility

Two groups of samples were identified, annealed, encapsulated and labeled. The groups were subjected to two different process cycles. A process cycle was identified as annealing, irradiation and readout, whereas the other process cycle excluded annealing. The samples were irradiated to a test dose of 4 Gy under standard irradiation conditions using a Cs-137 gamma cell. Thermoluminescence measurements were obtained after 24 h to eliminate the low-temperature peaks associated with shallow traps using the optimal heating rate. The process was repeated for seven cycles for both groups to estimate the effect of annealing on the sensitivity of the dosimeters.

### 2.2.6. Signal Stability

A group of samples was identified, annealed, encapsulated and labeled. The samples were irradiated to a test dose of 10 Gy under standard irradiation conditions using a Cs-137 gamma cell. Signal stability was estimated under two different storage conditions, i.e., a light-tight environment and visible light environment at room temperature. Thermal fading

was estimated by storing the samples in a light-tight environment at room temperature. Optical fading was estimated under two different storage conditions, i.e., under direct sunlight and under fluorescent light. Thermoluminescence measurements were obtained using an optimal heating rate.

### 2.2.7. TL Sensitivity

Two groups of dosimeters, synthesized glass and standard TLD-100, were identified, annealed, encapsulated, labeled and irradiated to a test dose of 5 Gy under standard irradiation conditions using a Cs-137 gamma cell. The absolute sensitivity can be determined using Equation (1) [18].

$$S = \frac{TL}{D \cdot m} \tag{1}$$

where *TL* is the thermoluminescent dosimeter response in reader unit, *D* is the test dose in Gy and *m* is the mass of the dosimeter.

The relative sensitivity can be determined by normalizing the TL response of the material of interest to that of the standard TLD-100 (LiF:Mg,Ti), given as Equation (2) [28]:

$$S(D)_r = \frac{S(D)_{TLD\ material}}{S(D)_{TLD-100}} \tag{2}$$

where $S(D)_{TLD\ material}$ and $S(D)_{TLD-100}$ are the absolute sensitivity of the test material and the standard TLD-100 (LiF:Mg,Ti), respectively.

### 2.2.8. Minimum Detectable Dose

The minimum detectable dose was calculated via the approach proposed by Pagonis and colleagues, given as Equation (3) [29]:

$$MDD = 3\sigma_{BKG}CF \tag{3}$$

where $\sigma_{BKG}$ represents the standard deviation of zero-dose measurements and *CF* is the calibration factor determined using Equation (4):

$$CF = \frac{D_c}{\frac{1}{N}\sum_{i=1}^{N}(M_i - M_{oi})} \tag{4}$$

where $D_c$ is the calibration dose, *N* is the number of TL dosimeters, $M_i$ is the measurement of the $i^{th}$ TL dosimeter and $M_{oi}$ is the background measurement of the $i^{th}$ TL dosimeter.

## 3. Results and Discussion

### 3.1. Sample Characterization

#### 3.1.1. Phase Analysis

Figure 1 presents a typical XRD spectrum of the synthesized $MgB_4O_7$:Zn glass. The amorphous nature of the synthesized sample is confirmed through the absence of sharp peaks and the presence of broad humps. The broad humps between 24° to 36° and 37° to 48° could be attributed to MgO and BO, respectively [26]. The two broad peaks in the XRD spectra result from short-range orders and confirm the amorphous nature of the synthesized samples [30]. The results demonstrate that $MgB_4O_7$:Zn glass could be successfully prepared using the melt-quenching method.

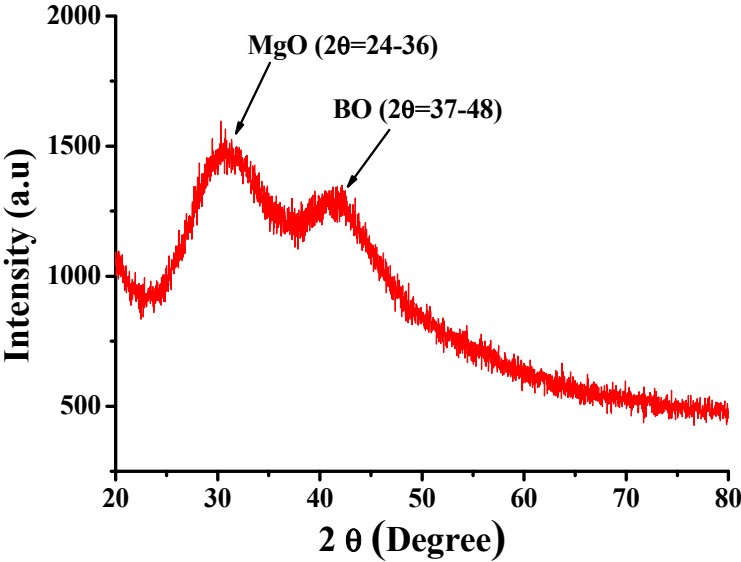

**Figure 1.** Typical XRD spectrum of MgB$_4$O$_7$:Zn glass.

### 3.1.2. Surface Morphology

A homogeneous amorphous surface morphology is confirmed through FESEM analysis. The absence of grains, as shown in Figure 2, further confirms the glassy nature of the synthesized samples.

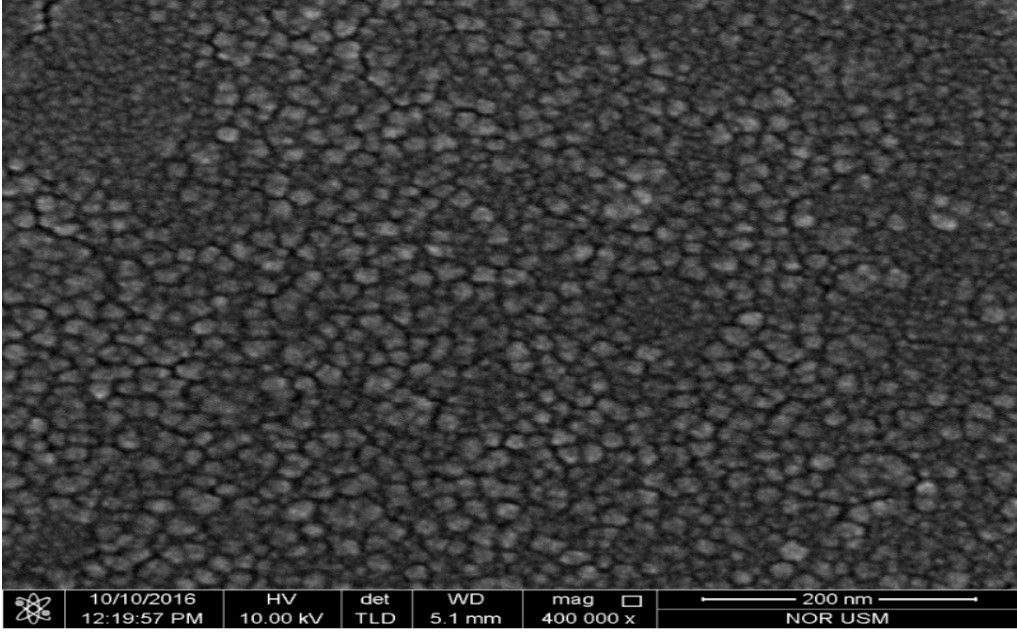

**Figure 2.** Surface morphology of MgB$_4$O$_7$:Zn glass observed by field emission scanning electron microscopy (FESEM).

### 3.1.3. Thermal Analysis

The glass-forming ability and glass thermal stability are the parameters of interest that demonstrate the ability of the proposed melt to resist crystallization when subjected to temperature variations. The characteristic temperatures, glass transition temperature, $T_g$, crystallization temperature, $T_c$ and melting temperature, $T_m$, used to estimate the glass-forming ability and glass thermal stability are identified from thermal analysis. Thermal events, endothermic or exothermic reactions, occur upon heating the glass and are identified through DSC [31]. The DSC thermogram of quenched MgB$_4$O$_7$:Zn glass, identifying

the thermal events of endothermic reaction (phase transition and melting) and exothermic reaction (crystallization), is depicted in Figure 3. The glass-forming ability, $T_{rg}$, was estimated using the Kauzamann relation, given as Equation (5) [32]:

$$T_{rg} = \frac{T_g}{T_m} \tag{5}$$

where $T_g$ is the glass transition temperature and $T_m$ is the glass melting temperature.

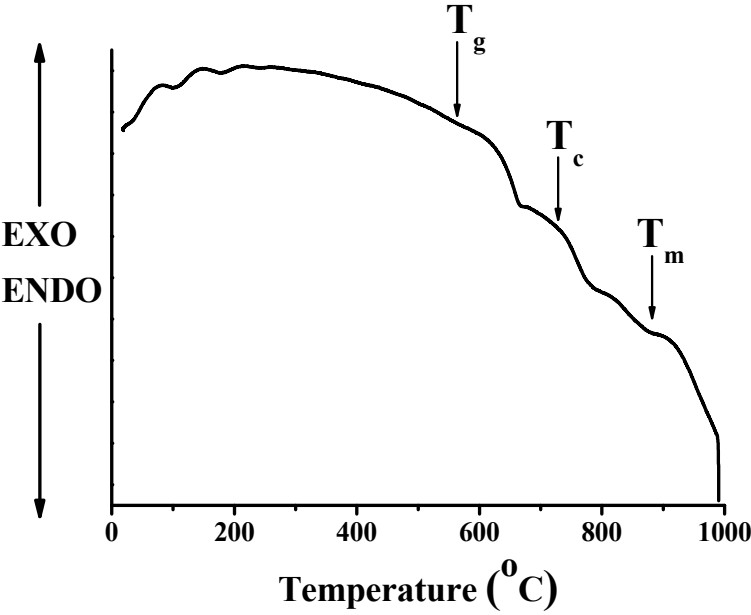

**Figure 3.** Differential scanning calorimetry (DSC) thermogram of S35654.

The glass thermal stability, $H_g$, resistance to nucleation upon heating, combining the nucleation and phase transformation growth aspects, was estimated using Hurby's assumption given as Equation (6) [33]:

$$H_g = \frac{T_c - T_g}{T_m - T_c} \tag{6}$$

where $T_g$ is glass transition temperature, $T_m$ is glass melting and $T_c$ is nucleation temperature. The smaller $T_m - T_c$ values detain nucleated crystal growth, whereas, higher $T_c - T_g$ values delay the nucleation process.

The glass transition temperature, $T_g$, glass melting, $T_m$, and nucleation temperature, $T_g$, of the synthesized glass were identified as 540 °C, 721 °C and 882 °C, respectively. The proposed mixture demonstrated good glass-forming ability, 0.61, within the identified range, $0.5 \leq T_{rg} \leq 0.66$ [34]. Glasses with $H_g \leq 0.1$ are hard to fabricate, requiring elevated cooling rates, whereas glasses with $H_g \geq 0.5$ are relatively easy to fabricate with standard quenching rates. The synthesized glass demonstrates excellent glass thermal stability at 1.62 [33].

### 3.2. Dosimetric Characterization

### 3.2.1. Heating Rate Optimization

The heating rate has a prominent effect on TL measurements, as stimulation energy significantly influences the trapped electron population [18]. Therefore, it is highly recommended to optimize the heating rate before further exploring the TL dosimetric properties for newly developed TL material. The heating rate affects the glow curve differently for first-order and second-order kinetics. The heating rate increment results in decreased glow curve intensity, shifting the maximum peak temperature to higher temperatures along

with broadening of the full width at half maximum (FWHM) of the glow curves, for the first-order kinetics. However, for second-order kinetics, the glow curve intensity increases with the increasing heating rate, and consequently, the maximum peak temperature shift towards higher temperatures [35]. Ideally, the heating rate significantly increases the peak intensity with a shift in the maximum peak temperature towards a higher peak temperature. However, in this study, the heating rate could be observed significantly shifting the maximum peak temperature towards a higher temperature without out affecting the peak intensity, as shown in Figure 4; this behavior could be attributed to the inhomogeneous sample size. The heating rate dependence of the TL response is shown in Figure 5, where each data point represents an averaged measurement value of three samples to minimize random errors. The heating rate was optimized between 1 °C s$^{-1}$ and 15 °C s$^{-1}$ due to the declining trend of TL response in the identified range. Although the maximum TL response could be observed with 1 °C s$^{-1}$, the optimum heating rate was identified as 7 °C s$^{-1}$, based on an economical and practical approach, which produces a significant TL response with minimal standard deviation and maximum peak temperature in the ideal range.

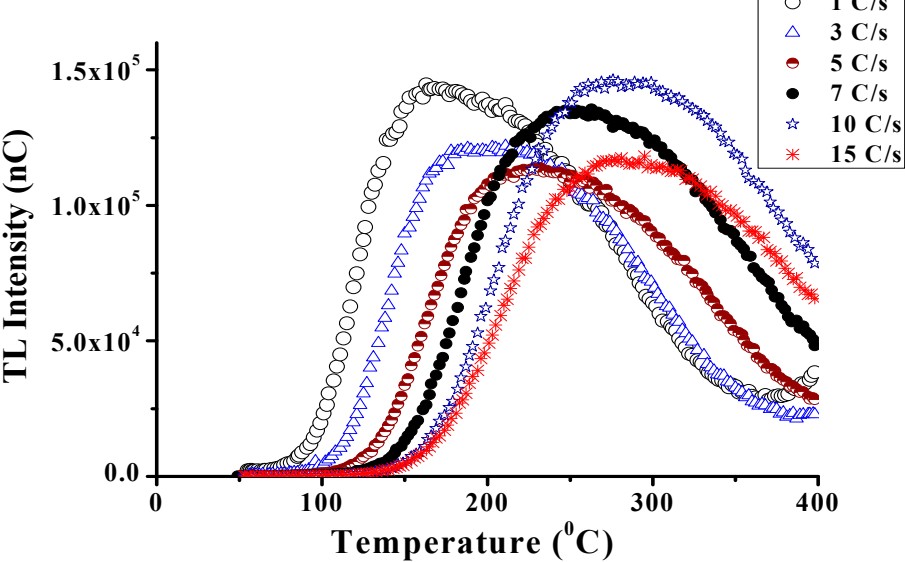

**Figure 4.** Glow curves of S35654 obtained with different heating rates.

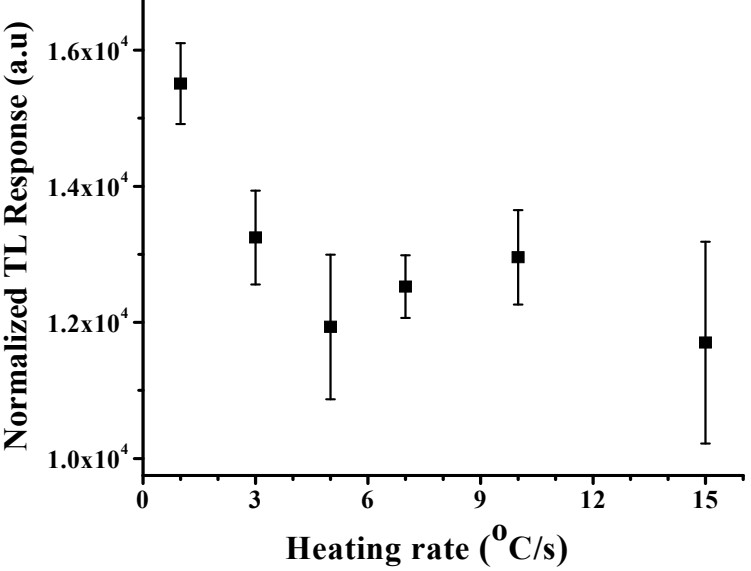

**Figure 5.** Heating rate dependence of thermoluminescence (TL) from MgB$_4$O$_7$:Zn glass.

### 3.2.2. Composition Optimization

Optimal glass composition was identified based on the dosimetric characteristics, i.e., TL response, maximum peak temperature and the effective atomic number of the synthesized composition. TL measurements were obtained using the optimal heating rate, $7\,^{\circ}\text{C s}^{-1}$. The glow peak intensity was observed to increase with the increase in the dopant concentration up to a nominal ratio of 0.4 mol%, which exhibited a decreasing trend afterwards. The effect of dopant concentration on the TL response of magnesium borate glass is evidenced in Figure 6. The inconsistency in the glow peak intensity can be attributed to the irregular sample shapes. The composition was optimized by normalizing the TL response and eliminating random errors by averaging multiple TL measurements for the individual data point. Figure 7 shows the normalized TL response of the synthesized $MgB_4O_7$:Zn glass series. The nominal 0.4 mol% ZnO was identified as the optimal concentration that produces a significant TL response with minimum standard deviation.

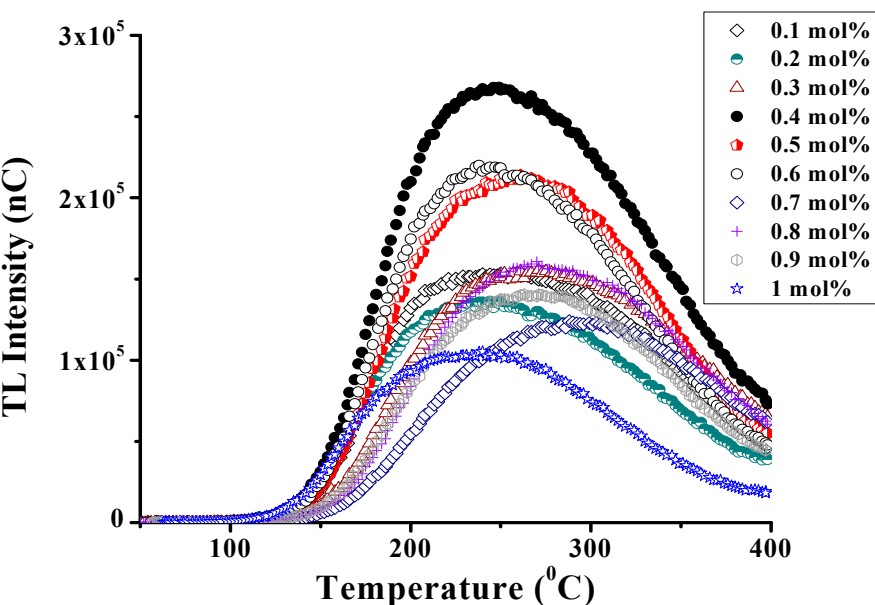

**Figure 6.** Glow curves of $MgB_4O_7$:Zn glass with different dopant concentrations.

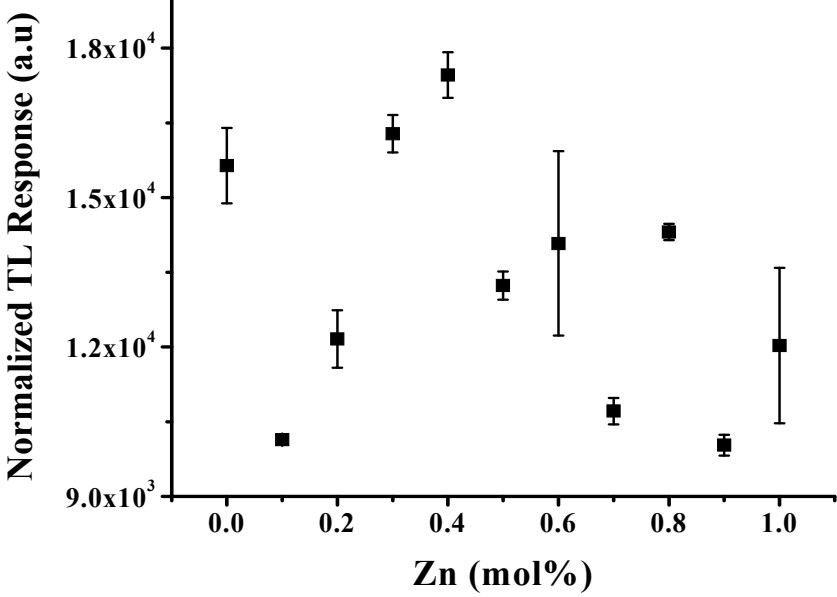

**Figure 7.** Effect of dopant concentration on TL response of $MgB_4O_7$:Zn glass.

### 3.2.3. Annealing Process

The effective deployment of newly synthesized TL material for radiation dosimetry requires a well-identified annealing procedure to resensitize the dosimeter by eliminating the irradiation history. Therefore, the annealing procedure was optimized by applying different thermal treatments by varying the annealing time and temperature. Figure 8 presents an annealing temperature dependence of S35654 (MgB$_4$O$_7$:Zn glass with 0.4 mol% ZnO). The annealing temperature was restricted to 400 °C to avoid thermal damage to the glass, which could potentially alter the intrinsic defects that could affect the dosimeter sensitivity. The annealing time dependence of the TL response is shown in Figure 9. The synthesized glass was observed to be significantly resensitized, producing a significant TL response with acceptable standard deviation, when subjected to 250 °C for 45 min. The identified annealing process is practically and economically feasible for routine clinical radiation dosimetry, as compared to the dual-step annealing process of standard TLD-100 that requires annealing the dosimeter at 400 °C for 1 h followed by 80 °C for 24 h [18]. Moreover, the sensitivity of standard TLD-100 was reported to be vulnerable to the annealing process [36]. Therefore, a well-defined and simple annealing process is always desirable to resensitize the dosimeter without affecting its sensitivity.

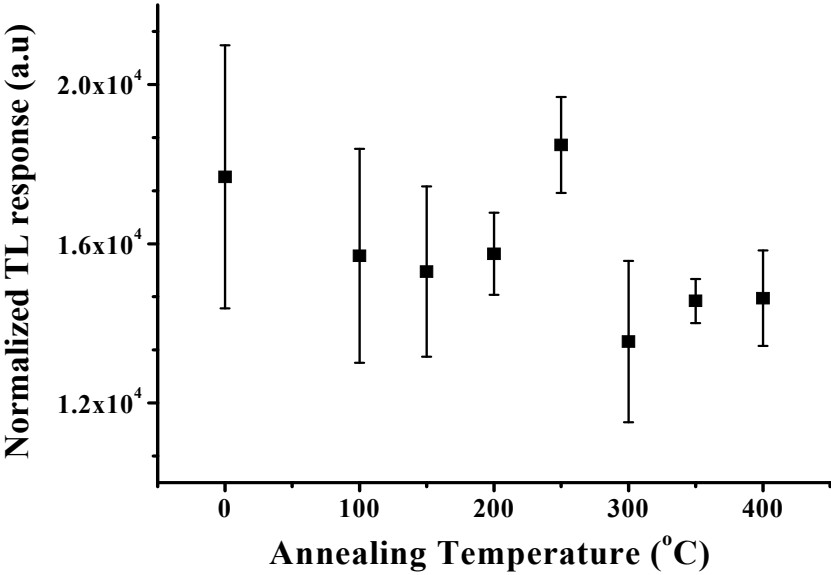

**Figure 8.** Effect of annealing temperature on TL response of MgB$_4$O$_7$:Zn glass S35654.

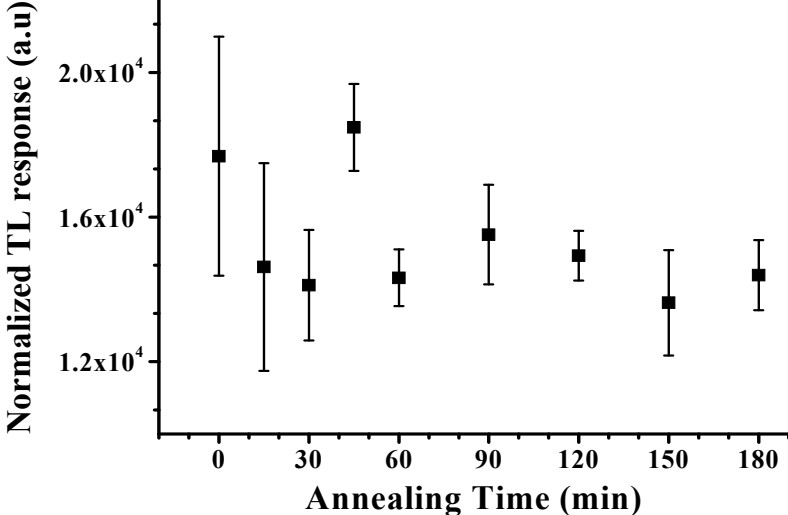

**Figure 9.** Effect of annealing time on TL response of MgB$_4$O$_7$:Zn glass S35654.

### 3.2.4. Dose Response

A linear dose response over a wide dose range is desirable for a dosimeter to be considered for multiple dosimetric applications. Figure 10 shows the dose response of MgB$_4$O$_7$:Zn glass irradiated with γ-rays; each data point represents the averaged measurement value of three samples. The dose response was observed to be linear throughout the tested dose range. The dose response is comparable with previously reported studies [9,16,37].

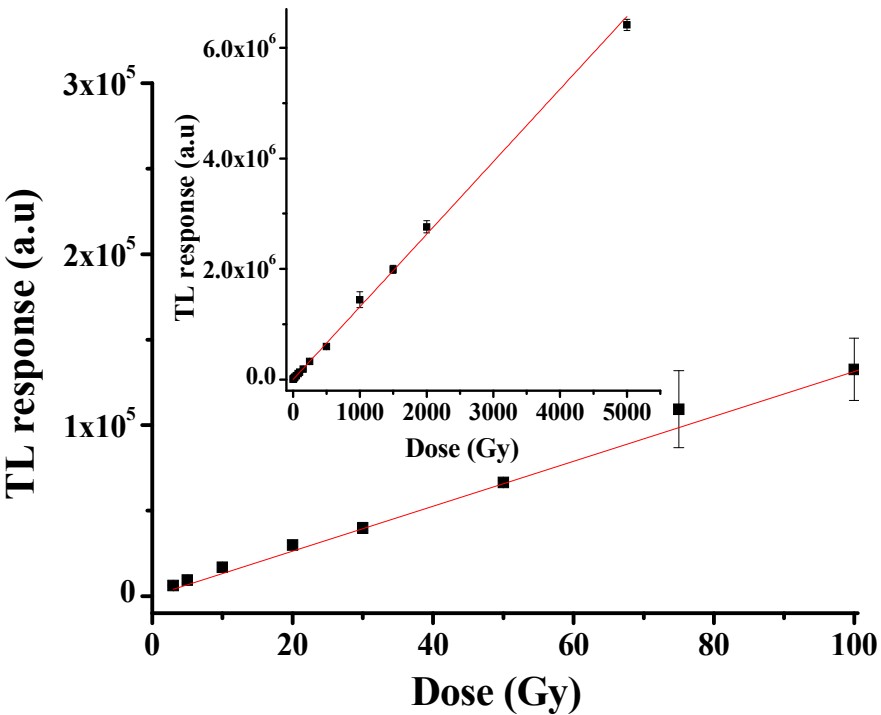

**Figure 10.** Dose response of MgB$_4$O$_7$:Zn glass S35654 for $^{137}$Cs source γ-rays.

### 3.2.5. Reproducibility

The high reproducibility of radiation dosimeters is important in clinical radiation dosimetry. The reproducibility is highly influenced by dosimeter homogeneity and irradiation history [38]. The reproducibility of the MgB$_4$O$_7$:Zn glass was assessed for seven consecutive cycles. The reproducibility test evaluates the thermal treatment applied to the dosimeters to resensitize the dosimeters, as well as to assess the homogeneity of the samples.

It can be evidenced from Figure 11, identifying radiotherapy dose measurement uncertainty limits, that both groups produce a similar trend over seven cycles, as the experimental procedures were carried out concurrently. The measurements were acquired with 6.8% uncertainty for annealed samples and 5.2% uncertainty for unannealed samples. The measurement uncertainties demonstrate that the dosimeter could be deployed for clinical dosimetry with proper dosimeter calibration. The maximum variation is slightly higher than the dose measurement uncertainty requirement, ±3.5%, for dose measurements in radiotherapy [39]. The fluctuation in the results could be attributed to the irregular shapes of the samples that affect the contact area between the sample surface and the heating planchet, resulting in a temperature gradient. Moreover, it was observed that the TL material could be resensitized adequately by read out at 350 °C, which is well above the optimum annealing temperature and is sufficient enough to de-trap the majority of the trapped electrons from their traps.

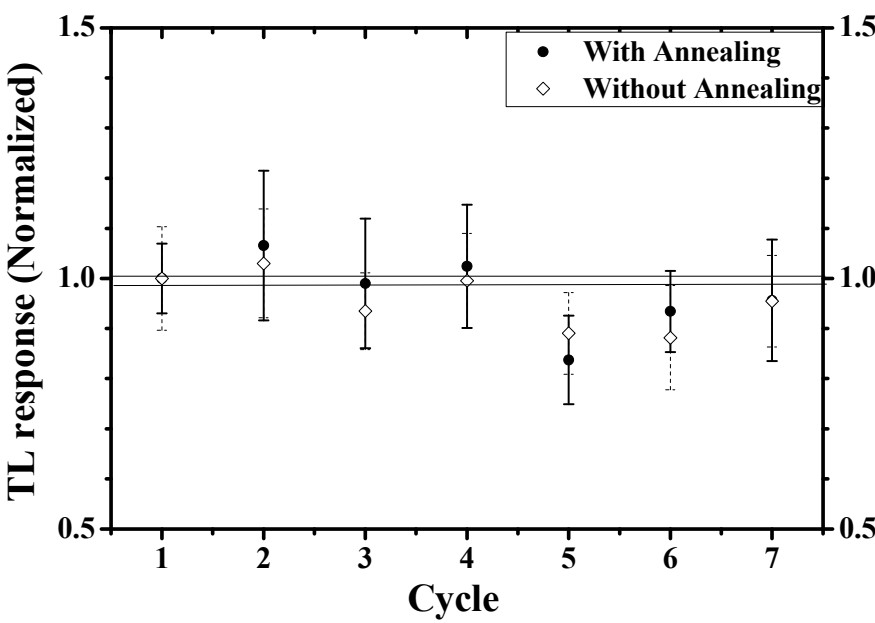

**Figure 11.** Reproducibility of $MgB_4O_7$:Zn glass S35654.

### 3.2.6. Signal Stability

Ideally, a TL material should retain dosimetric information over an indefinite course of time under normal storage conditions. However, some unfavorable circumstances disturb the signal stability, resulting in dosimetric information loss. Signal stability, also known as fading, is a statistical phenomenon in which trapped holes and electrons are released and consequently recombine, accelerating the depopulation of trapped electrons and holes. Signal stability is significantly important in personal and environmental radiation dosimetry where the dose is accumulated over a period. However, fading is not a major disadvantage for clinical dosimetry, as it could be overcome by storing the TLD under favorable conditions. Figure 12 shows the thermal fading of the synthesized glass. Each data point represents an averaged measurement value of three samples. Significant thermally induced signal loss, 18%, can be evidenced after a week, which subsequently increases up to 40% at the end of the second week. The light-induced fading, termed optical fading, behavior is shown in Figure 13. Most magnesium borate-based TL materials suffer from optical fading, adding considerable errors in the dose estimation [9,15]. The synthesized TL material was highly sensitive to visible light, resulting in drastic signal loss. The dosimetric information was observed to be reduced up to 60% after 1 h of storage under fluorescent light, whereas the signal was observed to be stable in sunlight for the first couple of hours, and then started to fade drastically. This behavior could be attributed to UV light, as the UV index was high during the first couple of hours of storage, whereas subsequent signal loss confirms the sensitivity to the visible light. Each data point represents an averaged measurement value of the three samples.

### 3.2.7. TL Sensitivity

Sensitivity is one of the most significant dosimetric aspects of a TLD material that remarkably influences the dosimeter's efficiency. Absolute sensitivity is significantly influenced by random errors introduced by the TLD reader using optical filters and thermal gradient due to the heating planchet as well as dosimeter-dependent errors, i.e., inhomogeneous defect concentration, dirt contamination and an irregular surface that introduces thermal gradient due to improper surface contact. Therefore, relative sensitivity was introduced to overcome the uncertainties associated with absolute sensitivity. The absolute sensitivity was observed to be 1.87 $\mu C \cdot Gy^{-1} \cdot g^{-1}$, whereas the relative sensitivity was observed to be 0.0176. The low relative sensitivity could be attributed to the defect population.

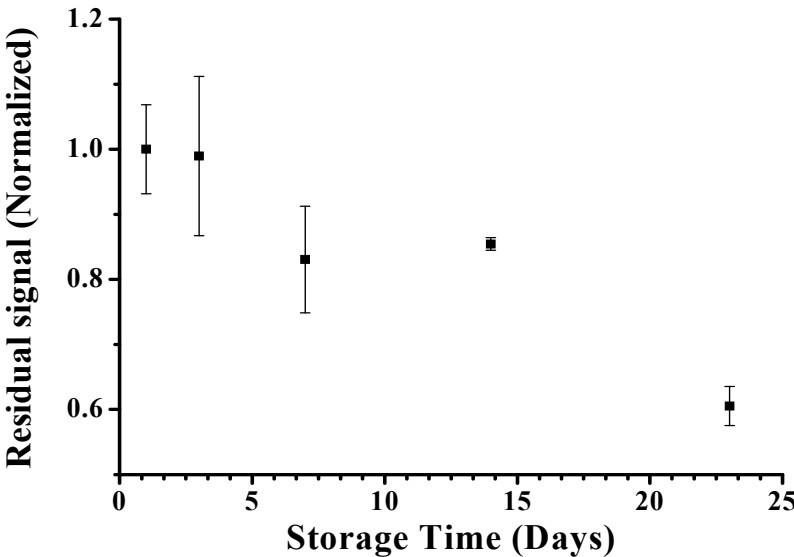

**Figure 12.** Thermal fading behavior of MgB$_4$O$_7$:Zn glass S35654.

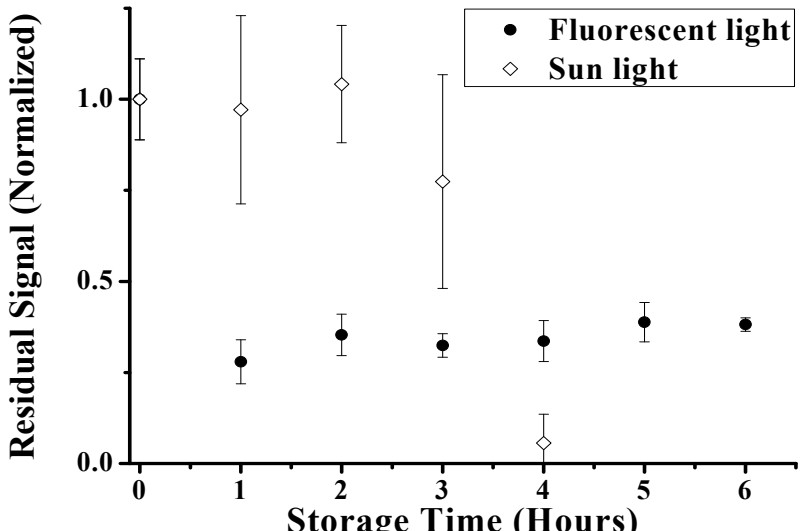

**Figure 13.** Optical fading behavior of MgB$_4$O$_7$:Zn glass S35654.

### 3.2.8. Minimum Detectable Dose

The minimum detectable dose or lowest detection limit is important in clinical, personal and environmental radiation dosimetry. The lowest detection limit is strongly influenced by the dosimeter composition, shape and size, as well as the TLD reader characteristics. The synthesized material was observed to detect doses as low as 6.16 mGy, which is comparable to previously reported studies, i.e., 0.01 mGy and 0.05 mGy [5,11]. The difference between the minimum detectable doses could be attributed to the defect population, which strongly depends on the nature of the synthesized material, i.e., crystalline or amorphous.

### 4. Conclusions

Original MgB$_4$O$_7$:Zn glass was synthesized successfully using the melt-quenching technique. It was confirmed that the MgB$_4$O$_7$ glass doped with 0.4 mol% of ZnO demonstrated promising dosimetric characteristics. The synthesized material revealed a simple glow curve with a single prominent peak at 240 °C with the optimum heating rate (7 °C s$^{-1}$). The reproducibility results suggest that the complex annealing procedures associated with standard TLD materials could be avoided for MgB$_4$O$_7$:Zn glass, enhancing its economic

feasibility. The wide linear dose range of the synthesized material makes it suitable for multiple dosimetric applications, including environmental, clinical and industrial dosimetry applications. It is expected that the synthesized material will also be utilized for volumetric dose verification in advanced radiotherapy techniques.

**Author Contributions:** Conceptualization, M.B. and I.S.M.; methodology, M.B.; formal analysis, M.B., I.S.M. and H.Y.; data curation, M.B.; writing—original draft preparation, M.B.; writing—review and editing, N.A., J.H.D.W. and H.Y.; supervision, I.S.M.; project administration, I.S.M. All authors have read and agreed to the published version of the manuscript.

**Funding:** This research was jointly funded by Hiroshima university through JSPS KAKENHI Grant Number 18KK0147 and the Division of Research and Innovation, Universiti Sains Malaysia, through the Bridging Research Grant Scheme with project reference code (BG0316), and account (304/PFIZIK/6316531).

**Acknowledgments:** Author (M.B.) is thankful to The World Academy of Sciences (TWAS) and Universiti Sains Malaysia (USM) for the financial support in the form of a TWAS-USM fellowship. The authors are also thankful to the Malaysian Nuclear Agency for providing gamma irradiation and TLD reader facilities throughout the research period.

**Conflicts of Interest:** The authors declare no conflict of interest.

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
