# Peer review of "Zinc-Doped Magnesium Borate Glass: A Potential Thermoluminescence Dosimeter for Extended Range of Dosimetric Applications"

_applsci, doi:10.3390/app12157491_

Round 1

Reviewer 1 Report

Thanks for the opportunity to review this paper on Zinc Doped Magnesium Borate Glass. 

I believe this paper is outstanding and I have no comments except a minor error in line 187. 

Wish the authors all the best.

Author Response

Respected reviewer, thank you for the valuable suggestions. The manuscript has been updated as suggested.

Reviewer 2 Report

Journal: Applied Sciences (ISSN 2076-3417).

Manuscript ID: applsci-1803011

Type: Article

Title: Zinc Doped Magnesium Borate Glass: A Potential Thermoluminescence Dosimeter for Extended Range of Dosimetric Applications.

Authors: Muhammad Bakhsh, Jeannie Hsiu Ding Wong, Nisar Ahmad, Hiroshi Yasuda, Iskandar Shahrim Mustafa.

a)           Figure 1 the author can write the phases of the compound.

b)          Figure 2 the author can calculate the distribution of the particle size of the materials.

c)           Refer to these refs. Very useful for the B2O3 and ZnO materials

DOI: https://doi.org/10.1039/C6RA15060H

DOI: https://doi.org/10.1007/s10854-017-6660-9

Best Regards

Author Response

Respected reviewer, thank you for your valuable suggestions.  

  1. The phase of the synthesized material is already mentioned in the figure caption.
  2. The particle size distribution is not of interest as the luminescence property is independent of the particle size.
  3. The referred literature is undoubtedly interedting thanks for sharing.

Reviewer 3 Report

This paper presents interesting and useful results on a radiation detector for multiple applications.

The text is very clear, and the results are well presented and adequately discussed.

I have just a few recommendations:

1. Page 6, line 4: Something strange appeared: Error! Reference source not found

2. Page 7, line 13: Please use s instead of S for second.

3. Page 7, Figure 4: Please correct at the inset the heating rate unit: °C/s. Do not use sec for second (SI of units).

4. Page 8, Figure 5: Please use s instead of sec.

5. Page 8, line 4: Please use s instead of S at 7°Cs-1 or 7°C/s.

6. Page 11, Figure 10: The figure is OK, but it is more common to present dose response results in logxlog format. This is just a suggestion.

7. References: Please have a look about the positioning of the author name initials and family names through the list, using the Guide for Authors.

Author Response

Respected reviewer, thank you for your valuable suggestions.

Response to the recommendations.

  1. The manuscript has been updated as suggested.
  2. The manuscript has been updated as suggested.
  3. The manuscript has been updated as suggested.
  4. The manuscript has been updated as suggested.
  5. The manuscript has been updated as suggested.
  6. Thank you for the suggestion.
  7. Confirmed.